

# How tank-mix adjuvant type and concentration influence the contact angle on wheat leaf surface

Yanhua Meng[1], Qiufang Wu[1,2], Hanxue Zhou[3] and Hongyan Hu[4]

[1] Anyang Institute of Technology, Anyang, Henan Province, China
[2] Anyang Wheat Breeding Engineering Research Centre Research Room, Anyang, Henan Province, China
[3] Anyang Quanfeng Biotechnology Co., Ltd, Anyang, Henan Province, China
[4] State Key Laboratory of Cotton Biology, Institute of Cotton Research, Chinese Academy of Agricultural Sciences, Anyang, Henan Province, China

Corresponding author
Hongyan Hu,
huhongyan1986@163.com

## ABSTRACT

Currently, the utilization of unmanned aerial vehicles (UAVs) for spraying pesticides is a prevalent issue in Asian countries. Improving the pesticide efficiency of UAV spraying is a major challenge for researchers. One of the factors that affect the efficiency is the wetting property of the spraying solutions on crop leaves. Tank-mix adjuvants, which can modify the wetting ability of the solutions, are often used for foliar application. However, different types and concentrations of tank-mix adjuvants may have different impacts on the wetting properties of droplets. In this article, we investigated the effects of four tank-mix adjuvants, Beidatong (BDT), Velezia Pro (VP), Nongjianfei (NJF), and Lieying (LY), on the dynamic contact angle (CA) values of droplets on the adaxial surface of wheat leaves. We measured the dynamic CA values of various concentrations of each adjuvant solution and determined the optimal concentrations based on the CA values, droplet spreading time, and cost. The results showed that adding any of the four adjuvants decreased the CA values, but the patterns of decrease varied among them. The CAs of BDT and VP solutions decreased slowly during the observation time (0–8.13 s), while those of NJF and LY solutions decreased rapidly throughout the observation period. According to the dynamic CA values of different concentrations, the optimal concentrations of BDT, VP, NJF, and LY for wheat field application were 12%, 16%, 6‰, and 0.3‰, respectively. Alkoxy-modified polytrisiloxane adjuvant (LY) could be recommended as an appropriate tank-mix adjuvant for wheat field application, considering spreading efficiency and cost. This study provides theoretical and practical guidance for selecting and optimizing tank-mix adjuvants for UAV spraying.

# INTRODUCTION

Crops have always suffered from the continuous invasion and attacks by pests, diseases and weeds, which would result in yield reduction. The application of pesticides is usually adopted to maintain crop output (*Matthews & Thomas, 2000*; *Zhu et al., 2019*). Foliage application of pesticides is one of the most efficient approaches to protect arable crops from the harmful damage of pests and diseases (*Jensen & Olesen, 2014*). The wetting of a

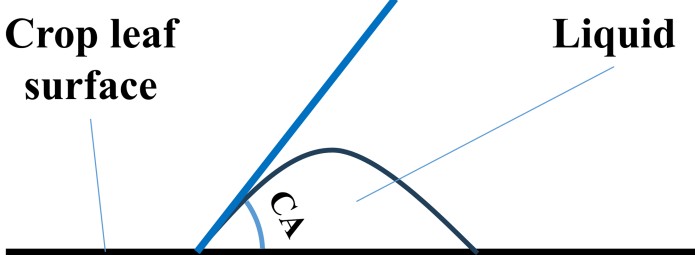

**Figure 1  Sketch map of CA.**

leaf with a pesticide solution depends on the properties of both the liquid and the solid substrate (*Quetzeri-Santiago, Castrejón-Pita & Castrejón-Pita, 2020*). Crop leaves are the main target of droplets in foliage application. The retention of droplets on crop leaves has a significant impact on pesticide efficacy (*Fang et al., 2019*; *Fountain, Harris & Cross, 2010*). The ability of a crop leaf to maintain water on its surface is regarded as leaf wettability (*Cavallaro et al., 2022*; *Fernandez et al., 2014*; *Papierowska et al., 2018*). The wettability of a crop leaf can be changed by the physicochemical properties of a liquid (*Da Silva Santos et al., 2021*; *Nairn, Forster & van Leeuwen, 2011*; *Sanyal, Bhowmik & Reddy, 2017*).

Contact angle (CA) is a measure of the wetting ability of a liquid on a solid surface, such as a crop (*Song et al., 2022*; *Wang et al., 2016*). In this study, CA refers specifically to the angle a liquid form between the interface of the leaf surface and liquid and the tangent to the liquid surface (Fig. 1). The smaller the CA, the higher the wetting ability of a liquid. A liquid that forms a CA smaller than 90° is categorized as a wetting liquid, while a liquid that forms a CA between 90° and 180° is a non-wetting liquid. A crop leaf with a CA below 90° is hydrophilic, while a crop leaf with a CA above 90° is hydrophobic (*Jeevahan et al., 2018*). A hydrophilic crop leaf allows the droplets to spread and evaporate quickly, while a hydrophobic crop leaf prevents the droplets from spreading and causes them to run off easily. Moreover, the non-spreading droplets take longer time to evaporate, which creates favorable conditions for plant pathogens to grow and spread (*Rowlandson et al., 2015*). Therefore, a wetting liquid is required to obtain a satisfactory biological control efficiency during a pesticide spraying application on a hydrophobic crop (*Meng et al., 2022*).

As mentioned above, the wettability of crop leaves is affected by the physicochemical properties of spray liquid, which directly influence the effectiveness of pesticides (*Sobiech et al., 2020*; *Zhang et al., 2017*). Tank-mix adjuvant can modify the physical and chemical properties of the spray liquid by lowering CA value and surface tension, reducing the negative effect of PH, increasing droplet size, and so on, which helps the spray liquid to spread on the crop leaf and enhance the efficiency of pesticides (*He et al., 2021*).

Normally, the nozzles of agricultural UAVs are at an altitude of 2 to 3 m above the crop canopy, while those of ground-based sprayer is at around 0.5 m above the crop canopy. This longer distance and the unpredictable crosswind increase the droplet drift potential (*Lou et al., 2018*). Furthermore, small droplet sizes, which are commonly seen in UAV spraying, also contribute to droplet drift (*Chen et al., 2020*). Thus, UAV spraying
pesticides are generally combined with the utilization of tank-mixed adjuvants to improve pesticide efficiency by reducing droplet drift (*Wang et al., 2018*; *Zhang & Xiong, 2021*). For a hydrophobic crop, the function of a tank-mix adjuvant is not only to reduce droplet drift but also to facilitate droplet spread on crop leaves as soon as possible (*Peirce et al., 2016*).

Wheat is a typical hydrophobic crop (*Song et al., 2022*). The CAs on the wheat leaves have been measured at 118–152° and 140–146° (*Marquez, Stuart-Williams & Farquhar, 2021*). Therefore, the wettability of pesticide solution is critical for controlling wheat diseases and pests. Several previous studies have used tank-mix adjuvants in the pesticide solution when applying aerial sprayers to enhance pesticide efficiency *Wang et al. (2022)* analyzed the droplet spectrum, drift potential index (DPI), field deposition, and control efficacy of different adjuvants on wheat rust and aphids. They found that the addition of adjuvants to the spray solution improved the control efficacy and duration of the pesticide. *Chen et al. (2018)* and *Zhang et al. (2018)* report that the tank-mix adjuvants can boost weed control efficiency in wheat fields. *Meng et al. (2018)* report that the use of tank-mix adjuvant can reduce imidacloprid dosage by 20% without increasing negative effects on wheat aphid control efficacy when using a UAV sprayer. *Wang et al. (2022)* explore that the addition of tank-mix adjuvants to spray solution can improve the control efficacy of wheat aphids and rust significantly and extend the duration of the pesticide. *Yan et al. (2021)* investigate that the addition of tank-mix adjuvant can improve the control effect of prothioconazole on Fusarium head blight in wheat and increase wheat yield. *Zhao et al. (2022)* report that the use of appropriate tank-mix adjuvants for UAV sprayers on wheat fields can significantly improve the performance of pesticides by increasing pesticide dosage delivery efficiency and disease control efficacy. They also explore that the use of tank-mix adjuvants can also help reduce the pesticide dosage while ensuring their effectiveness, which is similar to the conclusion of *Meng et al. (2018)* mentioned above. *Song et al. (2022)* evaluate four types of tank-mix adjuvants on wheat leaf by measuring metrics such as surface tension, CA, and so on, and the results indicate that the adjuvant type has a great effect on surface tension and CA value.

Although the effect of tank-mix adjuvants on the pesticide efficiency of wheat pests and disease control is explored widely, the measurement of dynamic CA values of different tank-mix adjuvants under a serial concentration is rarely reported. During the actual spraying process, the droplets that land on the crop leaves will gradually expand and flatten on the wheat surfaces, and their shapes will vary over time. This is especially true for liquids with tank-mix adjuvants, which exhibit more noticeable changes in their droplets. Static CA measurements are applicable for situations where the droplet movement or deformation is negligible and are mainly used to evaluate the wettability of a solid surface and the stability of a droplet on a solid surface. The dynamic CA is appropriate for measuring the CA of a moving droplet and is mainly used to investigate the dynamic behavior of a droplet rolling on a solid surface, the variation of wettability, and the stability of a droplet on a tilted surface (*Johnson, Dettre & Brandreth, 1977*). The measurement of dynamic CA can provide information about the dynamic response and kinetic behavior of droplets under different conditions, and the analysis of CA changes of droplets under different velocities, slopes, droplet deformations, *etc.*, can help to understand the dynamic properties of the

interface between droplets and solids. Therefore, the use of the dynamic CA in measuring the diffusion characteristics of droplets on wheat leaf surfaces can accurately reflect the wetting properties of droplets.

The main objective of this study was to investigate the influence of tank-mix adjuvant type and concentration on CA values on wheat leaf surface to select the appropriate adjuvant type and corresponding concentration for the control of wheat pests and disease when UAVs are adopted as sprayers.

## MATERIALS AND METHODS

### Materials

The variety of wheat used in this study was Zhoumai 22, which was planted on the campus experimental field of Anyang Institute of Technology. Wheat leaves were collected freshly during the late flowering period, a critical time for wheat pests and disease control.

The tank-mix adjuvants used in this study were Beidatong (BDT) (methylated plant oil, Hebei Mingshun Agricultural Co., Ltd, China), Velezia Pro (VP) (mineral oil, TotalEnergies Fluid company, Courbevoie, France), Nongjianfei (NJF) (hyperbranched fatty alcohol ether modified polymer, Guilin Jiqi Biochemical Co., Ltd, Guilin, China), and Lieying (LY) (alkoxy modified polytrisiloxane, Anyang Quanfeng Biotechnology Co., Ltd, Henan, China).

### CA value measurement

The laboratory experiment was designed to optimize the appropriate concentration of four tank-mix adjuvants (BDT, VP, NJF, and LY) by measuring dynamic CA values on the adaxial surface of wheat leaf under different concentrations, respectively. The four adjuvants were mixed with tap water as the tested aqueous solution with different concentrations, respectively. The dynamic CA measurement was not feasible for LY concentrations above 0.3‰ due to the rapid diffusion of droplets on the wheat leaf surfaces. Therefore, only LY concentrations below 0.3‰ were measured. Table 1 showed the concentration levels and measurement times of the four adjuvants used for the dynamic CA measurements.

The CA value of each concentration was measured on the adaxial surfaces of three freshly undamaged wheat leaves collected from the experimental field. Adhesive tape was adopted to fix the tested leaf on the glass slide (25 cm × 76 cm) to facilitate the capture of images for CA measurement. The interval of image capture was 0.07 s, and the dynamic CA value was measured from 0.00 to 8.13 s in most cases. The initial CA ($t = 0$ s) was recorded as $CA_{initial}$ and it was compared between solution concentrations of the same tank-mix adjuvant. The final CA (the last measuring time) was recorded as $CA_{final}$. The change in CA value was shown in the following equation.

$$CA_{change} = CA_{initial} - CA_{final} \tag{1}$$

The CA was measured by the sessile drop method using an optical tensiometer Attention Theta Flex (Biolin Scientific, Stockholm, Sweden) with a high-resolution camera (1,984 × 1,264 px with a maximum of 3009 FPS) and LED light. The details of the measuring process can be found in the previous study (*Meng et al., 2022*). The laboratory measurements were performed at a constant relative humidity of 57% and room temperature of 27 ± 0.4 °C.

**Table 1** Solution concentration of the adopted tank-mix adjuvants, and the corresponding observing time and number of CA.

| Adjuvant | Solution concentration | Observing time (s) | | Number of measured CAs of each solution concentration |
|---|---|---|---|---|
| | | $t_{initial}$ | $t_{final}$ | |
| BDT | 2%, 4%, 6%, 8%, 10%, 12%, 14%, 16% | 0 | 8.13 | 114 |
| VP | 4%, 8%, 12%, 16%, 20%, 24%,28%, 32% | 0 | 8.13 | 114 |
| NJF | 0.2‰, 0.4‰, 0.6‰, 0.8‰, 1‰ | 0 | 8.13 | 114 |
| | 2‰ | 0 | 3.10 | 44 |
| | 3‰ | 0 | 2.88 | 37 |
| | 4‰ | 0 | 2.88 | 37 |
| | 5‰ | 0 | 3.38 | 44 |
| | 6‰ | 0 | 2.30 | 33 |
| | 7‰ | 0 | 1.51 | 23 |
| | 8‰ | 0 | 2.95 | 42 |
| | 9‰ | 0 | 1.44 | 21 |
| | 10‰ | 0 | 1.01 | 15 |
| LY | 0.1‰ | 0 | 8.06 | 113 |
| | 0.2‰ | 0 | 3.82 | 54 |
| | 0.3‰ | 0 | 2.59 | 37 |

**Notes.**
Observing time $t_{initial}$ indicates the first measured CA, while $t_{final}$ is the last measured CA.

## Data processing and analysis

All data were statistically analyzed using software SPSS version 20.0 for Windows (SPSS Inc., Chicago, IL, USA), and Tukey's test was used to analyze differences between treatments at the 0.05 level of significance. Origin2021(Academic) (Origin Lab, Northampton, MA, USA) was adopted to draw the figures.

## RESULT

### Dynamic CA on wheat leaves of four tank-mix adjuvants

As shown in Figs. 2 and 3, a notable decrease in the CA values was observed after the addition of the four tank-mix adjuvants, respectively. CA value of tap water on wheat leaf adaxial surface was around 142.89°, which agreed with the result of the previous study (*Marquez, Stuart-Williams & Farquhar, 2021*). It could be seen that the CA behaviour of BDT and VP were similar (Fig. 2), while those of NJF and LY were alike in most measuring cases (Fig. 3). The appearance and shape of tap water droplets on wheat leaf adaxial surface during the observing time was shown in Supplemental Information 1.

In the case of BDT, the highest $CA_{initial}$ (90.63°) and $CA_{final}$ (69.53°) were observed for the concentration of 2%, and the lowest $CA_{initial}$ (60.04°) and $CA_{final}$ (44.56°) were observed for the concentration of 12% (Figs. 4A, 4B). In the low-concentration BDT group of 2%, 4%, 6%, and 8%, the $CA_{initial}$ decreased with the increase of concentration, but the CA values were similar after 0.3 s except for the concentration of 2%. In the
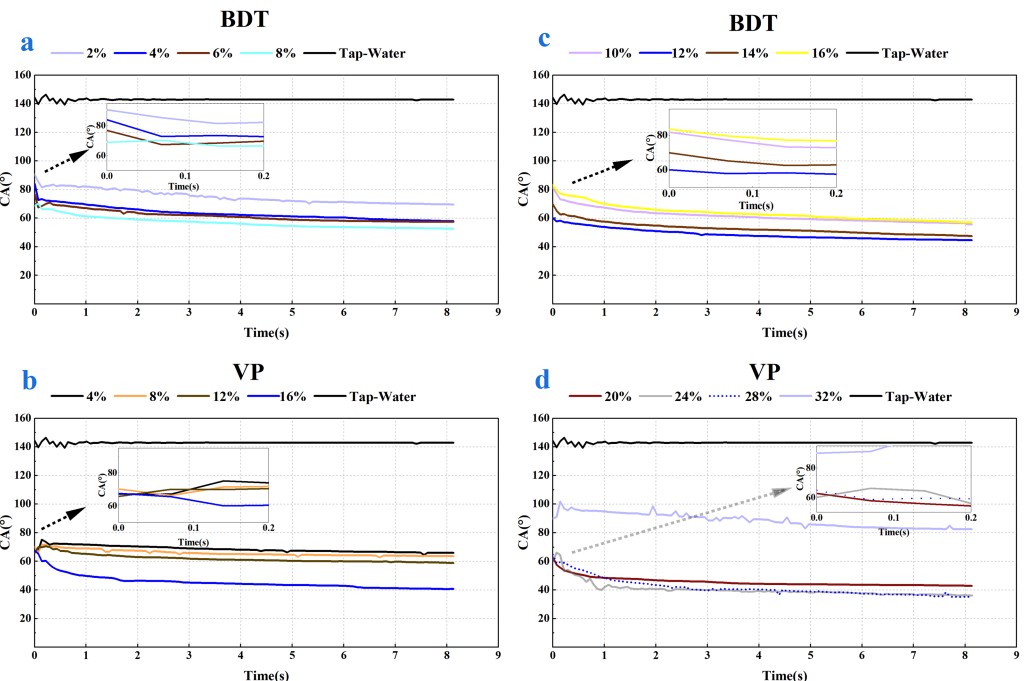

**Figure 2** **Dynamic CA of BDT and VP under different solution concentration, respectively.** (A) Contact angle changes over time after adding 2%–8% BDT tank-mix adjuvant. (B) Contact angle changes over time after adding 4%–16% VP tank-mix adjuvant. (C) Contact angle changes over time after adding 10%–16% BDT tank-mix adjuvant. (D) Contact angle changes over time after adding 20%–32% VP tank-mix adjuvant.

high-concentration BDT group of 10%, 12%, 14% and 16%, the highest $CA_{initial}$ value was found in a concentration of 10% (83.85°), followed by 16% (83.39°), 14% (69.82°), and 12% (60.04°). The expansion of the droplets on wheat leaves was shown in Supplemental Information 2.

In the case of VP, the highest $CA_{initial}$ was 89.99° (32%) and the lowest $CA_{initial}$ was 60.10° (24%) (Fig. 4C). The highest and the lowest $CA_{final}$ was 82.49° (32%) and 35.16° (28%), respectively (Fig. 4D). The CA values of concentration 32% decreased slightly over time but keep at above 80° over the whole observing time. CA values of concentration 16% dropped below 60° after 0.10 s and decreased slightly but stay above 40° during the remaining observing time (0.10–8.13 s). CA values of concentration 20%, 24%, and 28% were kept at around 36–45° after 3 s (Fig. 2D), while the CA values of the remaining concentrations were 58–68° after 3 s (Fig. 2B). The appearance shape of VP droplets dissipating on wheat leaf adaxial surface over 8.13 s was shown in Supplemental Information 3.

In the case of NJF, the $CA_{initial}$ of all concentrations was between 39–80° (Fig. 3). The lowest $CA_{initial}$ value was observed for a concentration of 7‰ (39.94°), and the highest initial CA value was observed for a concentration of 0.4‰ (79.16°) (Fig. 5A). Figure 5B showed the highest $CA_{final}$ was 33.79° (0.2‰) and the lowest was 6.79° (6‰). In the group of 0.2‰, 0.4‰, 0.6‰, 0.8‰, and 1‰, CA values of each concentration decreased slightly during the observing time (Fig. 3A). It took around 6 s for the CA values of concentration

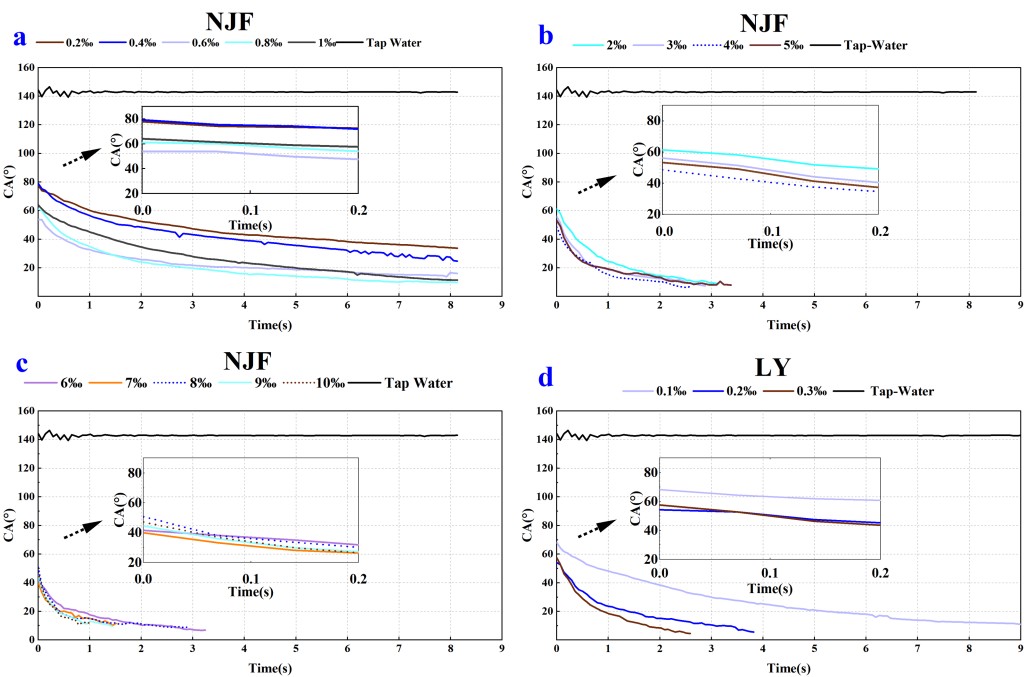

**Figure 3** **Dynamic CA of NJF (A, B and C) and LY (D) under different solution concentration, respectively.** (A) Contact angle changes over time after adding 0.2‰–1‰ NJF tank-mix adjuvant. (B) Contact angle changes over time after adding 2‰–5‰ NJF tank-mix adjuvant. (C) Contact angle changes over time after adding 6‰–10‰ NJF tank-mix adjuvant. (D) Contact angle changes over time after adding 0.1‰–0.3‰ LY tank-mix adjuvant.

0.2‰ and 0.4‰ to drop below 40°, but it only took 0.5s for CA values of concentration 0.6‰ and 0.8‰ to decrease below 40°. In the low concentration group of 2‰, 3‰, 4‰, and 5‰, the initial CA value was similar (48–61°) and CA values were below 20° after 1.6 s (Fig. 3B). In the high concentration group of 6‰, 7‰, 8‰, 9‰, and 10‰, the initial CA value was between 39–51° and CA values were below 20° in less than 1 s (Fig. 3C). The appearance shape of NJF droplets on wheat leaf adaxial surface over 8.13 s were shown in Supplemental Information 4. It could be seen that NJF droplet appearance shape changes notably on the wheat leaf adaxial surface under different concentrations.

In the case of LY, the $CA_{initial}$ of concentrations 0.1‰, 0.2‰ and 0.3‰ were 68.32°, 54.25° and 57.59° (Fig. 5C), respectively. It took around 5 s for the CA value of concentration 0.1‰ to decrease below 20°, but it only took less than 1s for the CA value of concentration 0.3‰ to drop below 20° (Fig. 3D). The highest $CA_{final}$ was 12.27° (0.1‰) and the lowest was 4.62° (0.3‰) (Fig. 5D). The highest $CA_{final}$ value was observed for concentration of 0.3‰ (4.62°) (Fig. 5D). The appearance shape of LY droplets on wheat leaf adaxial surface under three concentrations was shown in Supplemental Information 5.

## Decrease of CA

Figure 6 showed the decrease of $CA_{change}$ between different concentrations of the four adjuvants., respectively. In the case of BDT (Fig. 6A), the highest $CA_{change}$ was observed
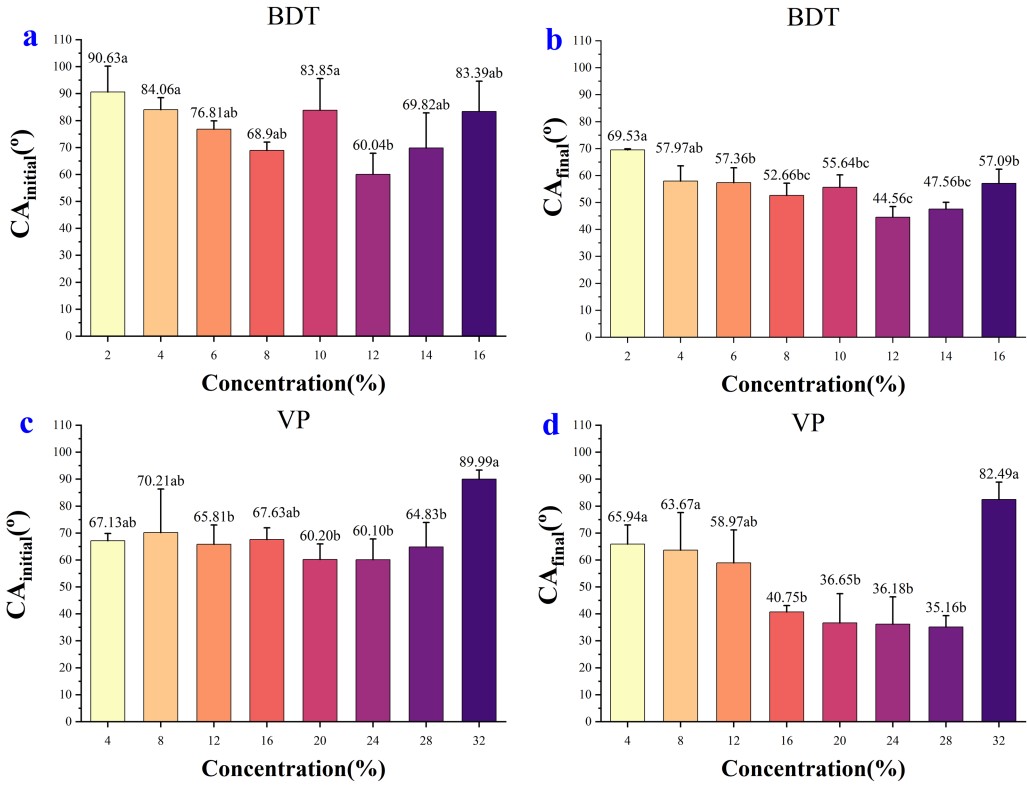

**Figure 4** **The initial CA and final CA of BDT and VP under different solution concentration.** (A) The initial CA after adding 2%–16% BDT tank-mix adjuvant. (B) The final CA after adding 2%–16% BDT tank-mix adjuvant. (C) The initial CA after adding 4%–32% VP tank-mix adjuvant. (D) The final CA after adding 4%–32% VP tank-mix adjuvant. Different lowercase letters indicate significant differences at the 0.05 level by Tukey's test.

in the concentrations of 10% (28.21°) and the lowest decreased in the concentrations of 12% (15.48°). Although the $CA_{change}$ between concentrations were observed in values, the differences between those decrease were negligible statistically. Thus, the ability of BDT concentrations to reduce CA on wheat leaf adaxial surface was similar based on the difference between the CA decrease.

In the case of VP (Fig. 6B), the highest decrease was seen in the concentrations of 28% (29.66°), and the lowest decrease was observed in the concentrations of 4% (1.19°). The low concentrations of VP had a weak effect on reducing CA, while the high concentrations had a strong effect, with the maximum effect at 28% concentration.

In the case of NJF (Fig. 6C), the highest $CA_{change}$ is 54.44° at 0.4‰ concentration and the lowest was 28.50° at 7‰ concentration. The $CA_{change}$ between concentrations was insignificant, except for the extreme values observed in the concentrations of 0.4‰ and 7‰.

In the case of LY (Fig. 6D), the $CA_{change}$ between concentrations was insignificant. The highest decrease was seen in the concentrations of 0.1‰ (56.05°), and the lowest decrease was investigated in the concentrations of 0.2‰ (48.64°).
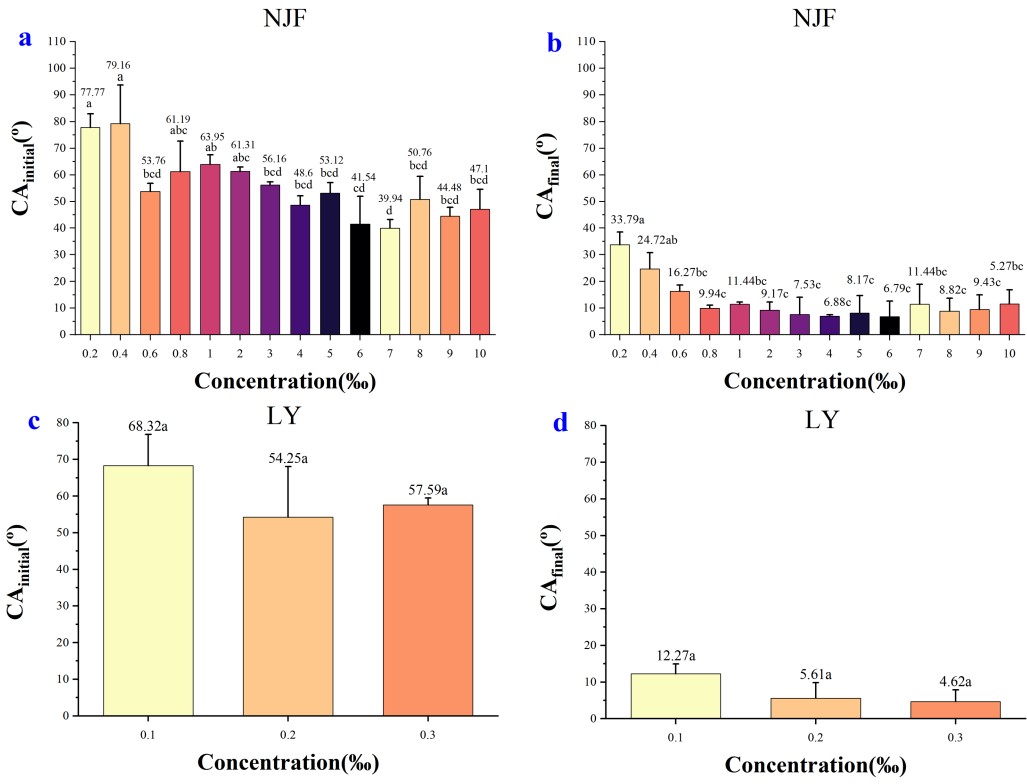

**Figure 5** **The initial CA and final CA of NJF and LY under different solution concentration.** (A) The initial CA after adding 0.2‰–10‰ NJF tank-mix adjuvant. (B) The final CA after adding 0.2‰–10‰ NJF tank-mix adjuvant. (C) The initial CA after adding 0.1‰–0.3‰ LY tank-mix adjuvant. (D) The final CA after adding 0.1‰–0.3‰ LY tank-mix adjuvant. Different lowercase letters indicate significant differences at the 0.05 level by Tukey's test.

## Optimal concentration selection

The optimal concentration for field spraying application depended on various factors, such as the $CA_{initial}$, $CA_{final}$, $CA_{change}$, diffusion time, evaporation rate, and product cost. Table 2 summarized the optimal concentrations of the four tank-mix adjuvants for each of the three scenarios of achieving the lowest $CA_{initial}$, lowest $CA_{final}$, and maximum $CA_{change}$.

In the case of BDT, the optimal concentration was 12‰, which resulted in the lowest $CA_{initial}$ and $CA_{final}$ among all concentrations and enabled rapid droplet spreading on the adaxial surface of the wheat leaf. Moreover, this concentration reduced the cost compared to higher concentrations.

In the case of VP, the optimal concentration was 16‰, which produced a similar $CA_{initial}$, $CA_{final}$, and diffusion time as the higher concentrations, but with a lower product cost. Although VP of 28‰ concentration had the largest CA reduction, it also increased the cost significantly.

In the case of NJF, the $CA_{initial}$, and $CA_{final}$ were lower when using concentrations of 6‰ and 7‰, and the time required for droplet to spread on wheat leaves was shorter. Compared to the 7‰ NJF tank-mix adjuvant, the addition of 6‰ concentration of

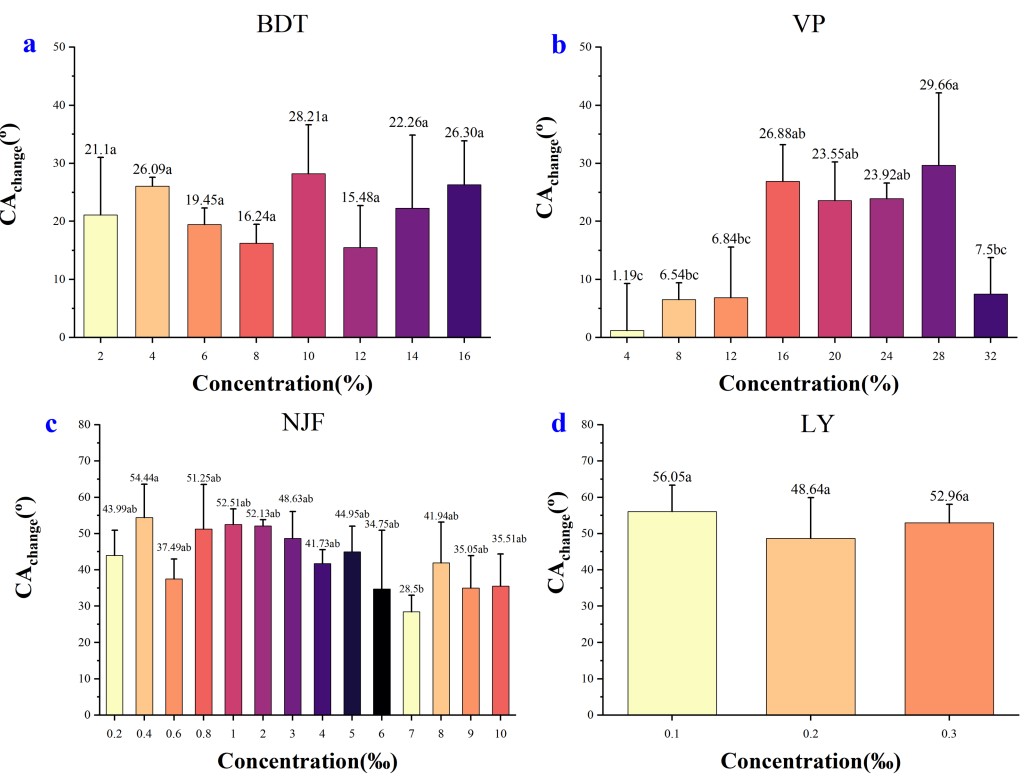

**Figure 6** Difference of $CA_{change}$ from the initial measuring time ($t_{initial}$) to final measuring time ($t_{final}$). (A) Change in CA value after adding 2%–16% BDT tank-mix adjuvant. (B) Change in CA after adding 4%–32% VP tank-mix adjuvant. (C) Change in CA after adding 0.2‰–10‰ NJF tank-mix adjuvant. (D) Change in CA after adding 0.1‰–0.3‰ LY tank-mix adjuvant. Different lowercase letters indicate significant differences at the 0.05 level by Tukey's test.

**Table 2** The criterion of appropriate concentration optimization.

| Adjuvant | BDT | | VP | | NJF | | LY | |
|---|---|---|---|---|---|---|---|---|
| Judgment Criterion | CA (°) | CC | CA (°) | CC | CA (°) | CC | CA (°) | CC |
| Lowest $CA_{initial}$ | 60.04 | 12% | 60.10 | 24% | 39.94 | 7‰ | 54.25 | 0.2‰ |
| Lowest $CA_{final}$ | 44.56 | 12% | 35.16 | 28% | 6.79 | 6‰ | 4.62 | 0.3‰ |
| Maximum $CA_{change}$ | 28.21 | 10% | 29.66 | 28% | 54.44 | 0.4‰ | 56.05 | 0.1‰ |

**Notes.**
CA means contact angle, while CC denotes the corresponding concentration. $CA_{change} = CA_{initial} - CA_{final}$.

NJF resulted in the lowest $CA_{final}$ and the optimal solution diffusion. Therefore, the recommended optimum concentration of NJF was 6‰.

In the case of LY, the optimal concentration was 0.3‰, which resulted in the lowest $CA_{final}$ and the highest $CA_{change}$ among all concentrations. Both the three concentrations of LY reduced the CA of droplets on wheat leaves rapidly, of which 0.3‰ LY concentration was more effective for the CA reduction.

## DISCUSSION

Tank-mix adjuvants could enhance the retention, diffusion, and wetting effects of droplets on crop leaves by mitigating the evaporation, drift, and rebound of the spray solution (*Preftakes et al., 2019*; *Ryckaert et al., 2008*; *Sijs & Bonn, 2020*). Wheat is a superhydrophobic crop that has a leaf structure that impeds the spreading and retention of droplets on its surface (*Dorr et al., 2015*). In this work, we investigated the effects of different types and concentrations of tank-mix adjuvants on the CA of droplets on wheat leaf surfaces. The results showed that adding tank-mix adjuvants to tap water significantly reduced the CA values and improved the diffusion performance of droplets. Different types of tank-mix adjuvants had different degrees of influence on the CA reduction and liquid diffusion. The concentration of tank-mix adjuvant was also a crucial factor that affected the CA values and diffusion of droplets on wheat leaf surfaces.

Tank-mix adjuvants based on surfactants had the ability to lower droplet surface tension (*Hazen, 2000*), which was a key parameter to characterize the physicochemical properties of droplets (*Arand et al., 2018*). The decrease of surface tension resulted in the reduction of droplets' CA and facilitated the spreading of droplets on solid surfaces. In this study, we recorded and analyzed the dynamic CA values of droplets on wheat leaf surfaces after adding adjuvants. The results indicated that all four types of tank-mix adjuvants lowered the CA of droplets on the wheat leaf surface but the lowering ability was different. The LY, an organosilicon alkoxy compound, had the most pronounced effect on reducing the CA of droplets. This was in line with previous studies that organosilicon adjuvants could substantially lower the surface tension of pesticide solutions and improve the spreading efficiency of pesticides (*Magor et al., 2023*; *Policello & Murphy, 1993*; *Zi et al., 2021*). Although NJF (hyperbranched fatty alcohol ether modified polymer) reduced the CAs in a short time as LY did but with a much higher concentration (6‰). BDT was a plant oil-based adjuvant that could reduce the CA and augment the wetting property of pesticides by lowering the surface tension of droplets and dissolving the wax layer and cuticle layer of plant leaves. *Xiao et al. (2019)* reported that plant oil-based adjuvants could significantly improve the droplet coverage and retention of defoliants in cotton leaves. *Yuan et al. (2019)* explored that the application of Green-peel orange essential oil (GOEO) as a spray adjuvant had great potential to enhance the deposition and penetration of pesticides on the leaf surface so that it would increase the pesticide utilization rate. VP was a mineral oil-based adjuvant, which had a similar effect as BDT and other plant oil-based adjuvants. A previous study showed that plant oil-based and mineral oil-based adjuvants could remarkably improve the droplet coverage and retention of pesticides on leaf surfaces (*Santos et al., 2019*). Our experiments also suggested that oil-based adjuvants could effectively lower the CA value of droplets, which would improve the efficiency of pesticides.

As mentioned above, the concentration of tank-mix adjuvant was an important factor that influenced the performance of pesticides. For NJF and LY, the CA value of droplets decreased fast and significantly at different concentrations. For oil-based adjuvants BDT and VP, within a certain concentration range, the CA declined gradually with

increasing concentration. Noteworthy, adjuvants with high concentrations may have negative effects on pesticide absorption (*Buick, Buchan & Field, 2006*). Both BDT and VP showed the phenomenon that the effect was worse at high concentrations than at lower concentrations. It might be due to the concentration of adjuvant solution reaching critical micelle concentration (CMC), which caused the droplet to produce micelle force and prevents the CA from decreasing (*Wang & Liu, 2007*). Therefore, in the actual spraying, the optimal concentration should be determined by considering the comprehensive factors such as $CA_{initial}$, $CA_{final}$, $CA_{change}$ of the droplets, and the degree of product cost. Further experiments on exploring the relationship of CMC of tank-mix adjuvant and CA on wheat leaves were suggested to be carried out in future work, aiming to obtain more reliable and accurate experimental results for practical application.

## CONCLUSIONS

In this article, we measured the effect of different concentrations of tank-mix adjuvants on droplet CA. We obtained the optimal concentration of BDT, VP, NJF, and LY for practical application by considering CA changes, droplet diffusion time, and other factors comprehensively. Firstly, we found that all concentrations of tank-mix adjuvants decreased CA values, with BDT and VP adjuvants showing slow dynamic CA changes over 0–8.13 s, while NJF and LY adjuvants exhibited rapid CA reductions over the observation time. Secondly, CA differences were observed among concentrations within the same adjuvant. The appropriate concentrations of the four adjuvants for wheat field application were 12% (BDT), 16%(VP), 6‰ (NJF), and 0.3‰ (LY) based on the CA dissipation time and CA values observed from indoor experiments. Finally, considering spreading efficiency and product cost, a low concentration of alkoxy-modified polytrisiloxane adjuvant (LY) reduced the CA rapidly to very low values and might be a suitable adjuvant for wheat fields.

In conclusion, we advise that CA values should be measured to optimize appropriate concentration for field application to obtain satisfactory biological control efficacy. Furthermore, not only the initial CA value is important when assessing the wettability of different liquids and optimizing the appropriate concentration for a specific liquid on the same crop leaf surface, but also what happens with the liquid drops over the observing time.

## ACKNOWLEDGEMENTS

We thank Mr. Xiaochao Liu, Mr. Yifan Zhang, and Mr. Xintao Du for their kind help for this work.

### Funding

This work was funded by the National Natural Science Foundation of China (no. 32201659). The funders had no role in study design, data collection and analysis, decision to publish, or preparation of the manuscript.

## Grant Disclosures

The following grant information was disclosed by the authors:
National Natural Science Foundation of China: 32201659.

## Competing Interests

Hanxue Zhou is employed by Anyang Quanfeng Biotechnology Co., Ltd.

## Author Contributions

- Yanhua Meng conceived and designed the experiments, performed the experiments, analyzed the data, prepared figures and/or tables, authored or reviewed drafts of the article, and approved the final draft.
- Qiufang Wu conceived and designed the experiments, performed the experiments, analyzed the data, authored or reviewed drafts of the article, and approved the final draft.
- Hanxue Zhou conceived and designed the experiments, performed the experiments, authored or reviewed drafts of the article, and approved the final draft.
- Hongyan Hu conceived and designed the experiments, analyzed the data, authored or reviewed drafts of the article, and approved the final draft.

## Data Availability

The raw measurements are available in the Supplemental Files.

## Supplemental Information

Supplemental information for this article can be found online at http://dx.doi.org/10.7717/peerj.16464#supplemental-information.

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
