# Peer review of "How tank-mix adjuvant type and concentration influence the contact angle on wheat leaf surface"

_PeerJ, doi:10.7717/peerj.16464_

## Round 0.1 · original submission · Minor Revisions

UAV is a prominent topic among pesticide application methods nowadays, but our knowledge of UAV spraying is limited compared to conventional field sprayers. Therefore, all data related to UAV spraying is significant because this info may provide background for new studies.

Your article needs linguistic revision. I suggest you should get help from one of your colleagues or our editing service to edit your article.

Please follow the suggestions of all reviewers. If you cannot accept one or more of their suggestions, you should explain the reason why you refused it.

**Language Note:** The Academic Editor has identified that the English language must be improved. PeerJ can provide language editing services - please contact us at copyediting@peerj.com for pricing (be sure to provide your manuscript number and title). Alternatively, you should make your own arrangements to improve the language quality and provide details in your response letter. – PeerJ Staff

Reviewer 1 ·

Basic reporting

The authors investigated how the contact angle (CA) to hydrophobic leaves of wheat changed by varying the type and concentration of tank-mix adjuvant in a time-coarse manner. The results showed that LY adjuvant was the most effective for use on wheat, which would be useful for optimizing the adjuvant to be used in UAV-based pest control.
However, I do not recommend acceptance of this manuscript for publication due to insufficient preparation of the manuscript (structure and description of method, result and interpretation) as pointed out below.

Major concerns:
The dynamic measurement of CA is interesting in and of itself, but the justification for it needs to be provided in the introduction. What is the drawback of traditional analysis (measurement at one point in time), why is it necessary to use dynamic measurement, what advantages does it offer, and what possible impact does it have on the choice of adjuvant? Moreover, the authors should state which values for CA should be emphasized and clarify how to select appropriate concentration based on the CA values.

There are errors in the description of some values of the measurements. For example, the CAinitial in l. 214 is 60.04°, not 69.53°, and the 2% CAfinal in Figure 9b is 69.53°, but in the text (l. 213) it is 66.53°. I have not checked all of the values, and such errors greatly affect the interpretation and credibility of the results, so all of the figures in the text and figures of the Result section should be checked.

The results section is redundant; since the CA decrease simply refers the difference between CAinitial and CAfinal, I think that including it in the content of the first subsection would improve readability and make it easier to discuss what the appropriate concentration is. Also, listing the top 3 measurements as found in l. 226–234 seems unnecessary and should be avoided.

Discussion may contain phenomena that occurred during the study with a detailed explanation. Comparisons with related studies or similar studies can also be made in this section. In the manuscript, many previous studies are only summarized and not compared with the results or findings obtained (l. 253–258, 265–268, 272–274, 276–286). This information should be included in the Introduction or Materials & Methods. In addition, l. 289–305 is redundant, as most of the information is described in the Results section. Overall, we recommend that Results and Discussion be written in a single section.

Experimental design

The description for statistical analysis methods is completely lacking; it should be included in a section under Materials & Methods. In the Results section, the test methods and p-values should be described (l. 215–217,222–224,229–230,234–235,240).

Validity of the findings

no comment

Additional comments

Minor concerns:
There is mention of UAV in the title and introduction, but I am not sure if this is necessary. Authors should clearly explain how UAV is involved in the analysis method (e.g., sprayed by a nozzle used for UAV) in the Materials & Methods section.

The English text needs to be proofread. For example, in several places there is no space immediately after a punctuation mark, or a hyphen is used to indicate a range of values instead of an em-dash.

l. 23 Spell out UAV for readers not familiar with the field.

l. 31 Spell out tank-mix adjuvant (BDT, DDE, NJF, LY).

l. 38 Use em-dashes, not hyphens. This should be done throughout the text.

l. 45 Use "UAV" instead of the word "drone”

l. 62–76 Should be listed as a separate paragraph.

l. 149 CAchange is defined but not used in the Results section; the CA decrease should be reworded to CAchange in “Decrease of CA” section of the Results and in some of the Figures.

l. 201–207 Since higher concentrations of LY appear to have better CA values, it is appropriate to measure the higher concentrations (>0.3‰) as well. If there is a reason not to perform the measurement, it should be stated in the text.

l. 216, 235 The word "notably" should not be used. The word "significantly" should be used when describing the presence or absence of statistical significance, and "insignificant" or "negligible" should be used when significance is not found.

l. 241 should clearly state why it is appropriate to set the concentration at 6‰. In CA decrease, 7‰ appears to be the best.

l.246–250 Although the Result section, Optical concentration selection, is an independent paragraph, there is no need to separate the sections because additional information are scarcely provided in the section .

l.326 It should be 0.3‰ (LY), not LY (0.3‰).

Figure 2,4,5,7,8 These should be moved to Supplemental files. This is because the appearance of droplets is not useful when examining differences between test intervals, although it would be useful for intuitive understanding. When interpreting the results, most of readers may refer to other Figures where quantitative values are given.

Figure 9–11 “CAchange” should be used." The meaning of "a", "b", and "c" should be indicated and what the error bars indicate (e.g., standard deviation, standard error, etc.) should be clarified. The numbers on the vertical axis of each figure should be the nice round number (20, 40, 60, 80, 100, 120 instead of 22, 44, 66, 88, 110).

Figure 10 NJF and LY, not BDT and VP.

Table 2 Not mentioned in the text. This table should be used as a basis for discussing the optimal concentration, as it would be helpful in selecting the optimal concentration.

Annotated reviews are not available for download in order to protect the identity of reviewers who chose to remain anonymous.

·

Basic reporting

In general article is understandable but still in some parts language is not clear enough. Also there are some grammer mistakes that should be revised.
Literature references are chosen releated to the subject. It has been used fairly enough number of literature.But it can be added more literature on UAV spraying in wheat with adjuvant effects.
The article is structured in an acceptable format. It has all the necessary sections required by the journal. Figures and tables are given seperately. They are labelled correctly.Since there are differences in horizontal and vertical spacing in some graphs, the evaluation may be misleading for the reader. For this reason, the same range units should be used in graphs as much as possible.It may be better for flow if some figures follow each other (eg fig3 and fig6).

Experimental design

It’s an original research. It fits journal’s aims and scope.The study is designed on laboratory studies. There are some deficiencies in both the material part and the method part of the study.For example, the devices used in the measurements are not included in the material section. Similarly, there is no information on how the adjuvants are applied on the wheat leaf in the method section. In order for the study to be clear and understandable, the deficiencies should be completed. The data given in the table is sufficiently descriptive and does not need to be re-given in the paragraph (for example, data in table 1).

Validity of the findings

In the results section, appropriate titles were created and their explanations were included. Results of the study has been given in order. It should be noted that it is important to examine how the CA changes depending on the drop diameter.
Conclusions should include not only a summary of the work done, but also recommendations based on the results obtained. Based on the results of this study, recommendations for direct field applications may be misleading. Because the results of the measurements made under laboratory conditions and without using any operation data of the drone cannot be directly interpreted. In addition, since the values of the 4 adjuvants tested under laboratory conditions at some concentrations are close to each other, it should be reassessed whether more than one adjuvant can be recommended.

Additional comments

It is strongly recomended that a simple field test should be added to laboratory test to check the data taken from laboratory tests.

·

Basic reporting

Text with very long paragraphs, which sometimes even discourages reading. There is a paragraph of almost a page. I advise better to distribute them. There are spelling errors, clearly evidenced, on line 214, by the word “concentration”.

Experimental design

I work on a topic relevant to the research area, presenting good methodology and good results, which can contribute as a source of more information about the studied problem.

Validity of the findings

The topic RESULTS and the topic DISCUSSION present some repetitive information. This is because the author made some discussions in the RESULTS topic, including citing BIBLIOGRAPHICAL REFERENCES. Therefore, I suggest that you better adapt the topics.

Additional comments

Some KEYWORDS are repeated in the TITLE, which is not adequate, as it limits the search for your possible future article by other researchers. I advise you to change the keywords to others that are not contained in the TITLE.

The CONCLUSION topic is not as direct as it should be, it seems more like “Final Considerations”. There is, for example, even an explanation of the methodology in this topic. I advise simplifying more and being more direct with what was concluded with the research.

---

## Round 0.2 · accepted · Accept

I would like to thank you for accepting the referees' suggestions and improving your article based on their suggestions. I think your article is ready to publish. We look forward to your next article.